# G2PTL: A Pre-trained Model for Delivery Address and its Applications in Logistics System

## Abstract

Text-based delivery addresses, as the data foundation for logistics systems, contain abundant and crucial location information. How to effectively encode the delivery address is a core task to boost the performance of downstream tasks in the logistics system. Pre-trained Models (PTMs) designed for Natural Language Process (NLP) have emerged as the dominant tools for encoding semantic information in text. Though promising, those NLP-based PTMs fall short of encoding geographic knowledge in the delivery address, which considerably trims down the performance of delivery-related tasks in logistic systems such as Cainiao. To tackle the above problem, we propose a domain-specific pre-trained model, named G2PTL, a **G**eography-**G**raph **P**re-**t**rained model for delivery address in **L**ogistics field. G2PTL combines the semantic learning capabilities of text pre-training with the geographical-relationship encoding abilities of graph modeling. Specifically, we first utilize real-world logistics delivery data to construct a large-scale heterogeneous graph of delivery addresses, which contains abundant geographic knowledge and delivery information. Then, G2PTL is pre-trained with subgraphs sampled from the heterogeneous graph. Comprehensive experiments are conducted to demonstrate the effectiveness of G2PTL through four downstream tasks in logistics systems on real-world datasets. G2PTL has been deployed in production in Cainiao's logistics system, which significantly improves the performance of delivery-related tasks. The code of G2PTL is available at https://huggingface.co/Cainiao-AI/G2PTL.

## 1 Introduction

Text-based addresses contain abundant semantic and geographical information, which is especially important in logistics systems. Making better use of the information in addresses has always been a key research direction in logistics because it not only directly determines the output of address perception tasks such as address completion, address standardization, but also further affects the performance of various downstream tasks such as address parsing Li et al. (2019), geocoding Wing & Baldridge (2011a), logistics dispatching Lan et al. (2020), estimated time of arrival Gao et al. (2021), package delivery route prediction Wen et al. (2021). These tasks are advantageous to improving the efficiency of logistic services and reducing the burden on related practitioners.

Nevertheless, there are two problems in existing works: a) In current logistics systems, it is a common practice to encode addresses through geocoding service Goldberg et al. (2007), which converts an address to its correlated coordinates (i.e., latitude and longitude). However, geocoding ignores the internal semantic and geographical information in addresses, which can not satisfy the need of address perception tasks such as address completion and standardization, let alone the high-level downstream tasks. b) Other general methods based on natural language processing (NLP) such as word segmentation Cai & Zhao (2016) and named entity recognition Liu et al. (2022) cannot achieve promising results due to the lack of geographical knowledge in training corpora.

Pre-trained models (PTMs) provide a new way for the representation of natural language. By using large-scale raw text for model training, the model can learn a general representation of text Kenton & Toutanova (2019); Floridi & Chiriatti (2020); Sun et al. (2021). In Cainiao's logistics system, addresses are the input for many downstream logistics tasks. Despite initial gains that were obtained by applying generic PTMs to logistics-related tasks at Cainiao, a clear performance plateau over

time was observed. One of the main reasons for this plateau is the lack of readily geographic and logistics delivery knowledge in generic PTMs. To address this problem, we believe that a domain pre-trained model for logistics which containing geographic and logistics delivery knowledge is necessary. At the same time, relevant work in other fields (such as biomedical and academic research) Gu et al. (2021); raj Kanakarajan et al. (2021); Beltagy et al. (2019) has proven that domain-specific PTMs trained using domain-specific training corpora and domain-specific training tasks can significantly improve the performance of downstream tasks in the corresponding field. We believe that the pre-trained model in the logistics field should have the following three geographical understanding abilities: a) The ability to understand address semantics. b) The mapping ability of addresses and their corresponding geographic coordinates (longitude and latitude). c) The ability to understand the spatial topological relationships between addresses. However, existing geographic-related PTMs can not meet with all these abilities at the same time Gao et al. (2022); Huang et al. (2022); Ding et al. (2023), and the main research fields of these PTMs are map retrieval or urban space research, which are obviously different from the logistics field in terms of research purposes and methods.

Given the above, we propose G2PTL, a delivery address pre-trained model in the logistics field, which has the above three important capabilities at the same time. In the daily production of Cainiao's logistics system, we verified the effectiveness of our model. Specifically, we tried to design and improve from the following three parts: a) Dataset. We use Cainiao's logistics delivery data to build a large-scale delivery address heterogeneous graph for sampling training samples. We use the delivery address as the node of the graph, and use the package delivery record and location information as the edges of the graph. The connection relationship between nodes and edges in the graph represents the delivery knowledge and spatial relationship between addresses. Each training sample is a subgraph, which is part of the heterogeneous graph. By using subgraphs for training, we can better model the spatial relationships between addresses. b) Model structure. In our dataset, we use a heterogeneous graph to represent the spatial relationship between different addresses. Therefore, we choose to use a graph learning based model to learn the spatial relationships between addresses. G2PTL combines graph learning with text pre-training. G2PTL uses a Transformer encoder to learn text representations and uses a Graphormer encoder Ying et al. (2021) to learn graph information. c) Pre-training tasks. We used three different pre-training tasks to enhance the geographical understanding ability of G2PTL. We use the masked language modeling (MLM) task to improve the ability to understand address semantics of G2PTL. The geocoding task can help model learn the mapping relationship between addresses and geographic coordinates. At the same time, we design the Hierarchical Text Classification (HTC) Yu et al. (2022); Zhou et al. (2020) task to learn the administrative hierarchy information of geographical entities, and add constraints to the output of the model through the administrative hierarchy.

We evaluate G2PTL on four downstream tasks in a well-known logistics system Cainiao. Experimental results show that G2PTL achieves a significant improvement over other general NLP-based PTMs and geospatial representation PTMs when applied to logistics downstream tasks. At the same time, we also conducted ablation experiments to prove the importance of each module in G2PTL. Our contributions can be summarized in the following points:

- We propose a pre-trained model G2PTL for the delivery address to improve the performance of related tasks in the logistics field. To the best of our knowledge, this is the first attempt to pre-train the address for the logistics field.

- We construct a large-scale heterogeneous graph base on Cainiao's real-world delivery behaviors and then introduce the graph learning method and design text-based pre-training tasks to learn the unique spatial properties of the delivery address. The results of the ablation experiments demonstrate the importance of the graph learning module.

- Extensive experiments of downstream tasks, conducted on large-scale, real-world datasets, demonstrate the superiority and effectiveness of G2PTL. The successful deployment of G2PTL at Cainiao's logistics system has greatly improved the performance of logistics-related tasks.

## 2 G2PTL

In this section, we introduce the design and application details of G2PTL, which includes the following parts: the construction of dataset, model architecture, and pre-training tasks.

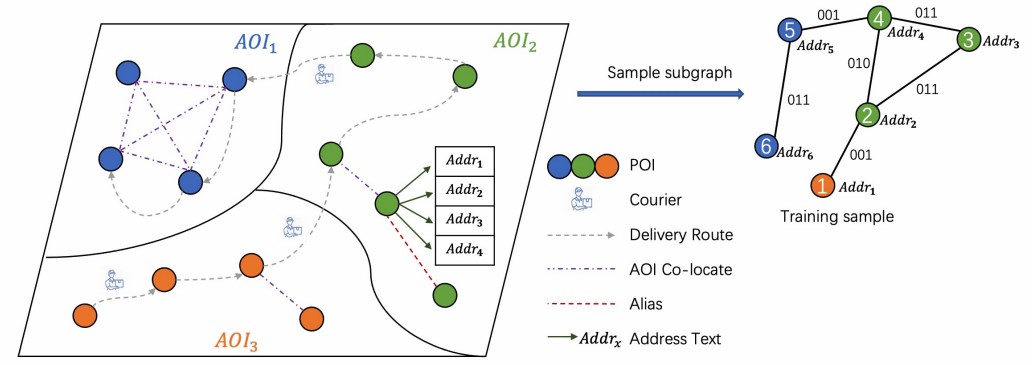

Figure 1: Heterogeneous graph of address and a training sample.

In the beginning, we introduce the definition of Area of interest (AOI) in the logistics field. AOI is a geographic block unit, which is generated by cutting the real-world based on the road information and the delivery behaviors. It can be used to express single or multiple regional geographical entities in the map (such as universities, apartment communities, office buildings, shopping malls, etc.). An AOI usually contains multiple POIs. For example, there are many stores in a shopping mall. AOI is the smallest geographic block unit in the logistics system, and address-related operations in the logistics system are mainly based on AOI.

## 2.1 DATASET

We use Cainiao's delivery data to construct a large-scale heterogeneous graph, which contains delivery addresses and their spatial location relationship. The heterogeneous graph takes delivery addresses as nodes and the relationship between addresses as edges. We sample from the graph to get subgraphs as the pre-training dataset (see Appendix A). The graph structure of the dataset is shown in Figure 1.

### 2.1.1 NODE

The delivery address in the logistics system is usually composed of province, city, district, town, and detailed address. In daily application, users usually add some address information (road, POI name, house number, etc.) and other remark information (put it at the door, put it in the courier cabinet, put it in the post station, etc.) to the detailed address. We design a paradigm ([POI administrative address + POI name]) to normalize the delivery address. POI administrative addresses are composed of "province + city + district + town + road + road number" in sequence. We also use this paradigm as the address representation of nodes in the heterogeneous graph.

### 2.1.2 EDGE

There are three types of edges in the heterogeneous graph of the dataset.

- Delivery Route. It is used to represent the courier's delivery route in the real logistics system, and this edge is obtained by analyzing the courier's daily delivery records.
- AOI Co-locate. For the logistics system, packages are usually delivered according to the AOI area, and different POIs within the same AOI should have a high degree of spatial correlation.
- Alias. For the same POI, different users will use different POI names, such as "CCTV Headquarters", which some users call it as "Big Pants" because of its pants-building shape. Correctly identifying aliases can greatly reduce the error rate of address parsing. We expect G2PTL to learn the alias relationship of different addresses, so if a node and another node are in the alias relationship, we will build an Alias edge between the two nodes.

There may be multiple types of edges between different nodes, so we encode the edges between nodes according to their types. We use the binary encoding to get the type code of an edge: the Delivery Route edge is at position 0, the AOI Co-locate edge is at position 1 and the Alias edge is at position 2. For example, there are Delivery Route edges and Alias edges between node A and node B, the edge between node A and node B is coded as "101".

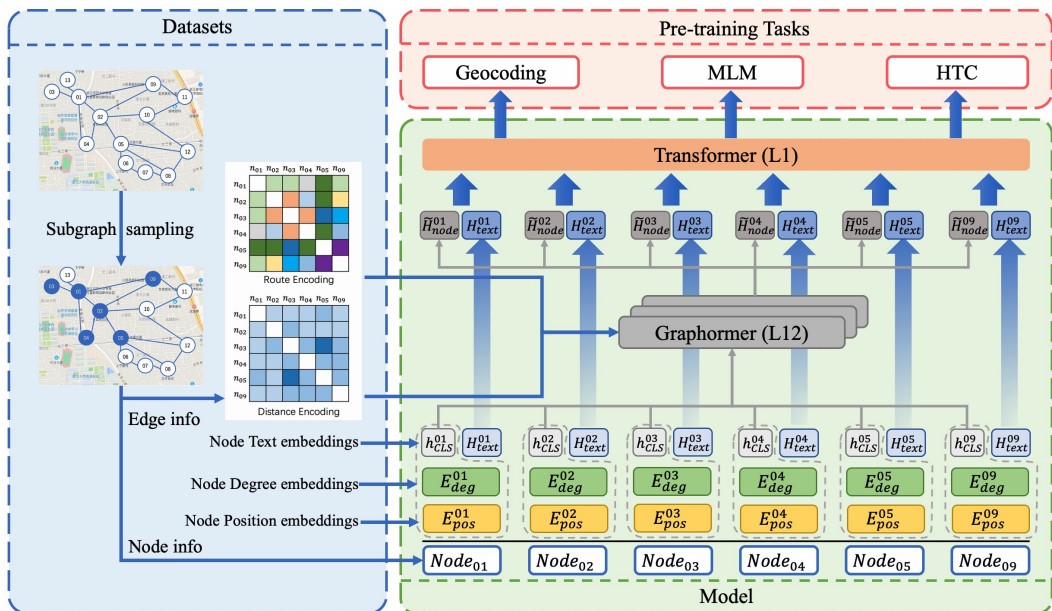

Figure 2: The architecture of G2PTL.

## 2.2 MODEL ARCHITECTURE

G2PTL is pretrained on the dataset mentioned above in a self-supervised manner. It consists of 3 parts: i) a Transformer encoder $Trans_{enc}$ for extracting the text representation of the address; ii) a Graphormer encoder for learning graph structure, which incorporates graph information by modifying the self-attention matrix based on the multi-head self-attention module; iii) a Transformer encoder $Trans_{pre}$ for aggregate and redistribute the information of each node in the graph. The model architecture is shown in Figure 2.

We first insert a "[CLS]" token into the head of the node address, and then use the sentence-piece algorithm to tokenize the text into a sub-word sequence $S_i = [s_i^{CLS}, s_i^1, s_i^2, ..., s_i^L]$. We use the $Trans_{enc}$ to get the text vector representation $\{h_{CLS}^i, H_{text}^i\}$ of node $v_i$:

$$\{h_{CLS}^i, H_{text}^i\} = Trans_{enc}(S_i) \tag{1}$$

where $h_{CLS}^i$ is the final hidden vector of the token "[CLS]", which is the aggregated representation of $S_i$, $H_{text}^i = [h_i^1, h_i^2, ..., h_i^L]$ and $h_i^1$ is the hidden vector of $s_i^1$. We use an embedding layer to encode the degree of the node to obtain its embedding vector $E_{deg}^i$. Similarly, we encode the position number of the node in train sample to obtain the position embedding vector $E_{pos}^i$. We can get the representation $H_{node}^i$ of node $v_i$:

$$H_{node}^i = h_{CLS}^i + E_{deg}^i + E_{pos}^i \tag{2}$$

The input of edge information in G2PTL depends on the node distance matrix and the node path type matrix. For a training sample with $n$ nodes, the node distance matrix records the shortest distance value between nodes, with dimension $[n \times n]$. The node path type matrix records the path information corresponding to the shortest distance between nodes, which is composed of the type code of each edge in the path, with dimension $[n \times n \times (n-1)]$. We use an embedding layer to encode the node distance matrix to obtain the *Distance Encoding* $E$ of the training sample; for the processing of the node path type matrix, we first use an embedding layer to encode it, and then average the embedding value in the node dimension to obtain the *Route Encoding* $R$ of the training sample. We add $E$ and $R$ as biases to the self-attention matrix of the Graphormer encoder.

To learn the spatial relationship between nodes, we first aggregate the representations of nodes and then use Graphormer encoder to redistribute it to each node. Specifically, we concatenate the representations $H_{node}^i$ of all nodes into a matrix $H = \{H_{node}^1, H_{node}^2, ..., H_{node}^n\}$, with dimension $[n \times d]$. The first Graphormer Layer in Graphormer encoder uses $H$ as input and outputs $H'$, $H'$

is also the input of the next Graphormer Layer. The calculation process of the Graphormer Layer is as follows:

$$Q = XW_Q, K = XW_K, V = XW_V \tag{3}$$

$$Attn(X) = softmax(\frac{QK^T}{\sqrt{d}} + E + R)V \tag{4}$$

$$X' = X + Attn(X), \widetilde{X} = X' + FC(X') \tag{5}$$

The input $X$ is projected by three matrices $W_Q \in \mathbb{R}^{d \times d}, W_K \in \mathbb{R}^{d \times d}, W_V \in \mathbb{R}^{d \times d}$ to the corresponding representations $Q, K, V$. $FC()$ contains 2 independent fully connected layers.

After computation of Graphormer encoder, $H$ is denoted as $\widetilde{H} = \{\widetilde{H}^1_{node}, \widetilde{H}^2_{node}, ..., \widetilde{H}^n_{node}\}$, $\widetilde{H}^i_{node}$ denotes the new representation with graph information of node $v_i$. We recombine $\widetilde{H}^i_{node}$ and $H^i_{text}$ of each node to form a new hidden vector representation of the node and use it as the input of $Trans_{pre}$:

$$\{\hat{H}^i_{node}, \hat{H}^i_{text}\} = Trans_{pre}(\widetilde{H}^i_{node}, H^i_{text}) \tag{6}$$

where $\hat{H}^i_{node}$ and $\hat{H}^i_{text}$ are used as input for pre-training tasks. In G2PTL, graphormer encoder takes the $H^i_{node}$ of each node as input, and obtains the output $\widetilde{H}^i_{node}$ of each node by aggregating and passing the features of all nodes. This will cause the information inside the $\widetilde{H}^i_{node}$ to be misaligned with the information in the $\hat{H}^i_{text}$ for each node, because the $\widetilde{H}^i_{node}$ already contains spatial topological relationship between different nodes and the aggregated knowledge of all nodes. So we use $Trans_{pre}$ to align the information in $\widetilde{H}^i_{node}$ and $\hat{H}^i_{text}$ for each node.

## 2.3 PRE-TRAINING TASKS

### 2.3.1 MASKED LANGUAGE MODELING

In logistics systems, the addresses provided by users are often incomplete, partially duplicated, or misspelled. We hope that when the address part is partially missing or incorrect, the model can still accurately identify the semantic information in the address. We use the MLM task to learn the semantic information in the address, and the input noise brought by the MLM task can enhance the model's ability to correct address errors when the address entered by the user is wrong. During the model training phase, we change the mask mechanism of the POI administrative address to the whole word mask (WWM), that is, once an administrative region is selected, all the words of the region name will be masked, each region has a 15% probability of being selected, and for the selected region, we replace the region name with "[MASK]" token with 80% probability, replace the region name with another region name of the same administrative level with 10% probability, and leave the region name unchanged with 10% probability. For the POI name, we use the same setting for mask probability, the difference is that the unit of the mask is each word in the POI name. This pre-training task enables the model to learn the relationship between POI names and the natural semantic information of address.

### 2.3.2 GEOCODING

In order to learn the relationship between address and geographic coordinates, we use the S2[1] geometry discrete global grid (DGG) system to map the coordinates of the address to its corresponding S2 cell (level 18), which covers an area around $1,200 m^2$. S2 system uses spherical projections, which significantly reduces the distortion caused by Mercator projection. At the same time, S2 is a hierarchical system in which each cell (we call this cell as S2 cell) is a "box reference" to a subset of cells. By using the S2 cell, the geocoding task is transformed into a classification task, and each S2 cell obtains the classification label using the 2Lt3C encoding method Huang et al. (2022).

### 2.3.3 HIERARCHICAL TEXT CLASSIFICATION

Delivery addresses are usually arranged from high-level administrative regions to low-level ones. In order to solve the problem of hierarchical structure of the address, we use the HTC task to learn the

---

[1]https://s2geometry.io

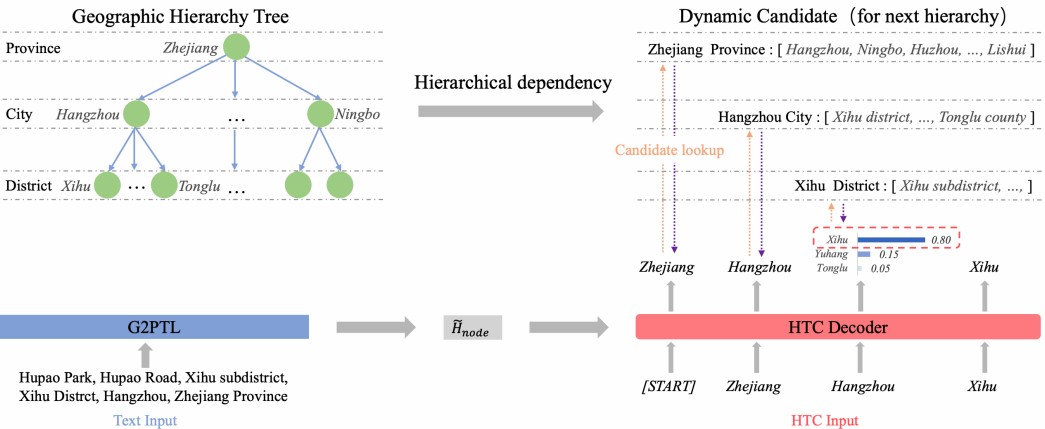

Figure 3: Illustration of the Hierarchical Text Classification task.

administrative hierarchical relations between different administrative regions. Recall that the goal of this task is to find a node path, which is from coarse-grained at the upper level to fine-grained at the lower level according to the level of the administrative region. We first construct a tree with a depth of 5 according to the hierarchical information of the administrative region, and each node of the tree represents an administrative region. According to the parent node and child nodes of the nodes in the tree, we can know the upper-level region information and lower-level region information of each administrative region. The model uses sequential forecasting to get the level of each administrative region in the delivery address. For each delivery address, we first forecast the administrative region whose level is 1 and then find out all its subordinate region sets in the tree, and the prediction results of the next level are constrained within this set until the last level.

## 3 EXPERIMENTS

In this section, we present the experimental results of G2PTL in different logistics-related tasks and demonstrate the superiority of G2PTL in logistics-related tasks by comparing it with other general NLP-based PTMs and geospatial representation PTMs. At the same time, we explore the importance of each module in G2PTL through ablation experiments.

### 3.1 LOGISTICS-RELATED TASKS

#### 3.1.1 GEOCODING

The geocoding task aims to convert the address into latitude-longitude coordinates of the corresponding location on earth. We use Cainiao's manually labeled data to evaluate the geocoding task and sample 80,000 addresses as the training set, 10,000 addresses as the validation set, and 10,000 addresses as the test set. For each address, we first use the 2Lt3C method to obtain its corresponding S2 cell (level 22) code representation. Then we use PTM to extract the hidden vector $h_{CLS}^i$ (see Equation 1) of the address as the input of the geocoding task, and use a fully connected layer for classification and fine-tuning. We use "Accuracy@N Km" as the metric, which measures the percentage of predicted locations that are apart with a distance of less than N Km to their actual physical locations.

#### 3.1.2 PICK-UP ESTIMATION TIME OF ARRIVAL

The Pick-up Estimation Time of Arrival (PETA) task aims to predict the arrival time of all packages not picked up by the courier in real-time, which is a multi-destination prediction problem without routing information. The improvement of PETA's accuracy rate can help the logistics system to better dispatch orders, measure the courier's overdue risk, and reduce customers' waiting anxiety. We use Acc@N M and Mean Absolute Error (MAE) as metrics for the PETA task. For a single package that has not been picked up, we calculate the minute difference k between the actual arrival time of the package and the model's estimated result. If k<N, the model's prediction for the package

is correct. We use FDNET Gao et al. (2021) as the backbone model to evaluate the PETA task, and we use $h^i_{CLS}$ of the PTM for the package address as input. We collected the delivery information of 170,000 real logistics packages and used them to generate 36,000 training samples, 5,000 validation samples, and 5,000 test samples.

### 3.1.3 ADDRESS ENTITY PREDICTION

The Address Entity Prediction (AEP) task aims to predict the missing administrative region information in delivery address. Many delivery addresses filled by users in the logistics system lack administrative region, and these incomplete addresses bring great challenges to address parsing service. In Cainiao, there are many orders that fail to be delivered every day, because the incomplete address cannot be parsed correctly. We collected the address data of 50,000 orders in Cainiao, and randomly masked the administrative region. The model needs to predict the masked part of the address. For the general PTMs, we first use PTM to extract the hidden vector $H^i_{text}$ (see Equation 1) of each word in the address, then use the fully connected layer for nonlinear transformation, and finally output the prediction results of N independent classifiers, where N is the length of the masked part. For G2PTL, we use the results of the HTC task as prediction results.

### 3.1.4 ADDRESS ENTITY TOKENIZATION

The Address entity tokenization (AET) task is similar to the named entity recognition (NER) task of NLP, which tokenizes the address into different geographic entities according to the administrative information of the entity. The accuracy of AET directly affects the performance of address parsing. The AET task mainly recognizes the contextual information of the word dimension in the address, and the model needs to classify each word in the address. We first use PTM to extract the hidden vector $H^i_{text}$ of each word in the address, then use the fully connected layer to perform the nonlinear transformation, and finally output the classification results of each word by N independent classifiers, where N is the length of the address.

At the same time, we use the public dataset GeoGLUE [2] to test different PTMs. Limited by the data format of GeoGLUE, we test AEP and AET tasks.

Table 1: Logistics-related Tasks and Datasets used to evaluate G2PTL.

| Task | #Train | #Val | #Test | Metric |
|------|--------|------|-------|--------|
| Geocoding | 80,000 | 10,000 | 10,000 | Acc@N km ↑ |
| Pick-up Estimation Time of Arrival | 36,000 | 5,000 | 5,000 | Acc@N M ↑ and MAE ↓ |
| Address Entity Prediction | 40,000 | 5,000 | 5,000 | Accuracy ↑ |
| Address Entity Tokenization | 40,000 | 5,000 | 5,000 | Accuracy ↑ |

## 3.2 EXPERIMENTAL SETUP

### 3.2.1 PRE-TRAINED MODELS

We first evaluate the performance of three widely used general PTMs in logistics related tasks, including BERT Kenton & Toutanova (2019), MacBERT Cui et al. (2020) and ERNIE 3.0 Sun et al. (2021). Considering the lack of geographical knowledge in the training corpora of general PTMs, we further test the performance of the geospatial representation PTMs in logistics related tasks, including GeoBERT Gao et al. (2022), MGeo Ding et al. (2023) and ERNIE-GeoL Huang et al. (2022). Since the data and code of ERNIE-GeoL are not released, we only adopt the pre-training objectives.

We also perform ablation experiments on multiple parts of G2PTL to determine their relative importance: a) G2PTL w/o GL: In this setting, we remove the Graphormer Layer of G2PTL. b) G2PTL w/o GT: In this setting, we remove the geocoding pre-training task. c) G2PTL w/o HTC: In this setting, we remove the HTC pre-training task.

The hyperparameters of $Trans_{enc}$ and $Trans_{pre}$ in G2PTL are the same as those of other PTMs, the hidden layer size is 768, and the number of attention heads is 12. The number of hidden layers in $Trans_{enc}$ is 12, and the number of hidden layers in $Trans_{pre}$ is 1. The Graphormer encoder

---

[2]https://modelscope.cn/datasets/damo/GeoGLUE/summary

Table 2: Comparison of pre-trained models on logistics-related tasks. "Avg." is the averaged score of the four tasks(no MAE).

| PTM | Geocoding | Pick-up Estimation Time of Arrival | | Address Entity Prediction | Address Entity Tokenization | Avg. |
|---|---|---|---|---|---|---|
| | Acc@1 km | Acc@20 M | MAE | Accuracy | Accuracy | |
| BERT | 54.45% | 17.88% | 69.84 | 49.56% | 84.13% | 0.515 |
| MacBERT | 52.15% | 16.64% | 70.60 | 50.26% | 84.15% | 0.508 |
| ERNIE 3.0 | 42.15% | 17.80% | 70.13 | 50.54% | 83.61% | 0.485 |
| GeoBERT | 50.30% | 18.16% | 69.84 | 52.42% | 89.89% | 0.527 |
| ERNIE-GeoL | 55.20% | 18.37% | 69.80 | 53.70% | 89.67% | 0.542 |
| MGeo | 60.40% | 18.30% | 70.37 | 53.28% | 90.98% | 0.557 |
| G2PTL | **65.95%** | **18.71%** | **69.38** | **91.94%** | **93.61%** | **0.676** |
| -w/o GL | 57.65% | 18.19% | 70.55 | 69.75% | 92.13% | 0.594 |
| -w/o GT | 54.95% | 18.30% | 70.31 | 89.14% | 91.44% | 0.635 |
| -w/o HTC | 55.45% | 17.68% | 70.84 | 58.58% | 90.96% | 0.557 |

of G2PTL contains 12 Graphormer Layers, of which the size of the hidden layer is 768, and the number of attention heads is 8. We train G2PTL for about a week on 10 Nvidia A100 GPUs. When fine-tuning on downstream tasks, we use the Adam optimizer Kingma & Ba (2014) with the learning rate initialized to $1 \times 10^{-6}$.

## 3.3 RESULTS AND ANALYSIS

We first evaluate the performance of G2PTL and other PTMs on logistics-related tasks. The results in Table 2 show that G2PTL significantly outperforms other PTMs on logistics-related tasks, and achieving the highest scores on each downstream task and the highest average score of 0.676. The results in Table 3 show that the address parsing performance of G2PTL is also better than other models on GeoGLUE Dataset. Experimental results show that G2PTL has comprehensively learned general geographic knowledge and logistics delivery knowledge, and this knowledge can significantly improve the performance of logistics-related tasks.

In the ablation experiments, we figure out the importance of each module in G2PTL. From the average score, the Geocoding task and the HTC task have the greatest impact on logistics-related tasks. Because G2PTL is a domain-specific PTM, its domain knowledge is mainly obtained through training corpora and pre-training tasks. Next, we explore the importance of each module of G2PTL in different downstream tasks. (1) For the geocoding task, we found that the "Acc@1 km" of "G2PTL w/o GT" has an absolute drop of 11% compared to G2PTL, which is also the largest drop in all ablation experiments. An important reason for this decrease is that geocoding is also a pre-training task because the "Acc@1 km" of "G2PTL w/o GT" is close to that of BERT. This also proves the importance of geocoding pre-training task. Com-

Table 3: Comparison of pre-trained models on GeoGLUE Dataset.

| PTM | Address Entity Prediction Accuracy | Address Entity Tokenization Accuracy |
|---|---|---|
| BERT | 49.97% | 92.47% |
| MacBERT | 48.63% | 91.93% |
| ERNIE 3.0 | 50.77% | 91.70% |
| GeoBERT | 52.23% | 94.02% |
| ERNIE-GeoL | 55.35% | 93.96% |
| MGeo | 56.93% | 94.57% |
| G2PTL | **74.07%** | **98.17%** |
| -w/o GL | 60.43% | 97.25% |
| -w/o GT | 68.87% | 96.48% |
| -w/o HTC | 57.33% | 94.12% |

pared with G2PTL, "G2PTL w/o GL" gets an absolute drop of 8.3%. This indicates that the introduction of the Graphormer to learn aggregation information between adjacent addresses can obtain better latitude and longitude mapping results. (2) For short-term metrics such as "Acc@20 Min" in the PETA task, the most important module is the HTC task, because these short-term metrics only focus on the address information of the current package, and do not care about long-term spatio-

temporal information. (3) For AEP and AET tasks, the most important module is also the HTC task, and the least important is the graph. On the one hand, the geographic administrative knowledge of the model is obtained through the HTC pre-training task, and on the other hand, AEP and AET tasks use the hidden vector of each sub-word in the input address, it focuses on the relationship of sub-words in the address, and does not focus on the spatial relationship between different addresses.

## 4 RELATED WORK

Domain-specific PTMs use the domain-specific corpora for pre-training and usually set MLM as a pre-training task, since most training corpora are text-based. Works such as BLURB Gu et al. (2021) and BioELECTRA raj Kanakarajan et al. (2021) in the biomedical field, PatentBERT Lee & Hsiang (2019) and SciBERT Beltagy et al. (2019) in the academic research field, FinBERT Araci (2019) in the financial field, ERNIE-GeoL and MGeo in map applications and BioBERT Lee et al. (2020), Clinical-BERT Alsentzer et al. (2019) and BlueBERT Peng et al. (2019) for healthcare have demonstrated the effectiveness of domain-specific PTMs on corresponding domain-related tasks.

When using deep learning models for geospatial representation learning, various types of spatial data are usually encoded into low-dimensional vectors in hidden space, that can be easily incorporated into deep learning models Mai et al. (2022). Wing and Baldridge explored the use of geodesic grids for geo-locating entire textual documents Wing & Baldridge (2011b). Speriosu and Baldridge propose a toponym resolution method that uses, as input to the language model, context windows composed of texts from each side of each toponym Speriosu & Baldridge (2013). Adams and McKenzie described a character-level approach based on convolutional neural networks for geocoding multilingual text Adams & McKenzie (2018). Cardoso et al. described a novel approach for toponym resolution with deep neural networks, which uses pre-trained contextual word embeddings (i.e., ELMo Peters et al. (2018) or BERT Kenton & Toutanova (2019)) and bidirectional Long Short-Term Memory units to improve the accuracy of geospatial coordinate prediction Cardoso et al. (2022). Word2vec proposed by Mikolov et al. Mikolov et al. (2013) is used to compute the POI-type embedding and model the relationship between the spatial distribution of POIs and land-use type Yao et al. (2017). However, Word2vec ignores statistical information and spatial information in geospatial data. To address this shortcoming, a series of related papers have been put forward continuously, such as POI2vec Feng et al. (2017), Region2vec Xiang (2020), Location2vec Zhu et al. (2019), GeoBERT, MGeo and ERNIE-GeoL. Gao et al. proposed a large-scale pre-training geospatial representation learning model called GeoBERT, which collected about 17 million POIs to construct pre-training corpora. MGeo represents the geographic context as a new modality and is able to fully extract multi-modal correlations for accurate query-POI matching. However, GeoBERT only modeled the address text, MGeo adds the latitude and longitude coordinates information to input, they do not consider the spatial topological relationship between addresses. Similar to our work, ERNIE-GeoL is trained on POI corpora, but it cannot make full utilization of the POI graph, since graph information of ERNIE-GeoL is introduced by the POI sequence, and the edge information of POIs in the graph is ignored.

## 5 CONCLUSIONS

In this paper, we proposed G2PTL, a pre-trained domain-specific model for delivery addresses in logistics field. G2PTL overcomes the shortcomings of other PTMs and boosts the performance of logistics-related tasks. With the specifically designed dataset, model architecture and pre-training tasks, G2PTL achieves excellent results in logistics-related tasks compared to other PTMs. G2PTL uses multiple encoder structures to learn the address representation with graph information, and utilizes three pre-training tasks to learn geographic knowledge and the delivery information of addresses. G2PTL is evaluated on four downstream tasks, which provide fundamental support to the logistics system. Experimental results and ablation studies demonstrate that G2PTL outperforms other general NLP-based PTMs and geospatial representation PTMs. The improvement brought by the deployment of G2PTL to Cainiao's logistics system proves that it can serve as a promising foundation for delivery-related tasks.

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

## A  TRAINING SAMPLE

The training sample of G2PTL is a subgraph sampled from the heterogeneous graph. We use $D[g_1, g_2, ..., g_n]$ to represent a training sample, where $g_i$ represents the node connection information in the heterogeneous graph. $g_i = (v_i, e_i, v_i')$, where $v_i$ and $v_i'$ represent two nodes connected by edge $e_i$. When sampling the data, we first set a base node $v_{base}$, and then sample a node from its first-order neighbor node set $V_{base}$. During the sampling process, the base node has a probability to be transferred to other nodes until the number of nodes in the sample $D$ reaches the set value. Finally, we obtain the mutual adjacency relationship between all nodes in $D$ from the heterogeneous graph to form the final training samples. Each training sample is finally represented as an independent subgraph.

In our experiments, we construct the heterogeneous graph using logistics delivery records within a 6-month from Cainiao. The heterogeneous graph contains 37.7 million address nodes, 108 million Delivery Route edges, 4.4 billion AOI Co-locate edges, and 3.1 million Alias edges. We sample 200 million samples from the heterogeneous graph, the number of nodes $k$ in each sample is set to 6, and the transition probability $p$ is set to 0.8. The data details for building heterogeneous graph are shown in Table 4.

Table 4: The data details for building heterogeneous graph. "Avg.Length" is the average length of delivery sequences.

| | |
|---|---|
| #Packages | 288.13M |
| #Addresses | 93.12M |
| #POIs | 31.64M |
| Time Range | 01/06/2022 - 16/11/2022 |
| #Citys | 355 |
| #Couriers | 0.55M |
| #Delivery sequences | 7.10M |
| Avg.Length | 37 |

---

**Algorithm 1:** Sampling train data on heterogeneous graph

---

**Input:** base node $v_{base}$, node number $k$ of sample, node transition probability $p$, heterogeneous graph $G$

**Output:** A training sample $D[g_1, g_2, ..., g_n]$

1   Initialize the node set $S_{node}$ of $D$, add $v_{base}$ to $S_{node}$;
2   Get the first-order neighbor node set $V_{base}$ of $v_{base}$ from $G$;
3   **while** $len(S_{node}) < k$ **do**
4     **if** $V_{base}$ *is not empty* **then**
5       Random sample a node $v$ from $V_{base}$;
6       Add $v$ to $S_{node}$;
7       Remove $v$ from $V_{base}$;
8       **if** *random() < p* **then**
9         Random sample a node $v'_{base}$ from $S_{node}$;
10        Get the first-order neighbor node set $V'_{base}$ of $v'_{base}$;
11        $v_{base} \leftarrow v'_{base}$, $V_{base} \leftarrow V'_{base}$;
12       **end**
13     **else**
14       Random sample a node $v'_{base}$ from $S_{node}$;
15       Get the first-order neighbor node set $V'_{base}$ of $v'_{base}$;
16       $v_{base} \leftarrow v'_{base}$, $V_{base} \leftarrow V'_{base}$;
17     **end**
18   **end**

---

## B   THE GEOGRAPHICAL KNOWLEDGE OF G2PTL

### B.1   GEOGRAPHICAL KNOWLEDGE OF RELATIVE LOCATIONS

To study the geographical knowledge of relative locations learned by G2PTL, we first use PTM to extract the hidden vector representation $h^i_{CLS}$ of the "[CLS]" token in the address, and then use the t-distributed stochastic neighbor embedding (t-SNE) method to project it into a two-dimensional space. We select seven districts in Hangzhou City, each district has 2,000 addresses. The t-SNE visualization results of BERT and G2PTL are shown in Figure 4. From this, we can see that the t-SNE embeddings predicted by G2PTL (Figure 4(b)) are more discriminative than those predicted by BERT (Figure 4(a)), with higher aggregation in the same regions. This shows that G2PTL can distinguish addresses located in different regions, and can learn geographical proximity information of different regions. For example, the XiHu district and the GongShu district are adjacent in Figure 4b, which is the same as in the real world. At the same time, we use Silhouette Score and Calinski-Harabaz (CH) Index to evaluate the sample aggregation degree of t-SNE visualization results, and the results are shown in Table 5. Compared with other general PTMs, G2PTL obtained significantly better aggregation evaluation results, and the CH Index reached 4,854.81, which shows that G2PTL has an excellent ability to identify address areas. The results of the ablation experiments in Table 5 prove that both the model structure and the pre-training tasks make significant contributions to G2PTL's geographical knowledge learning.

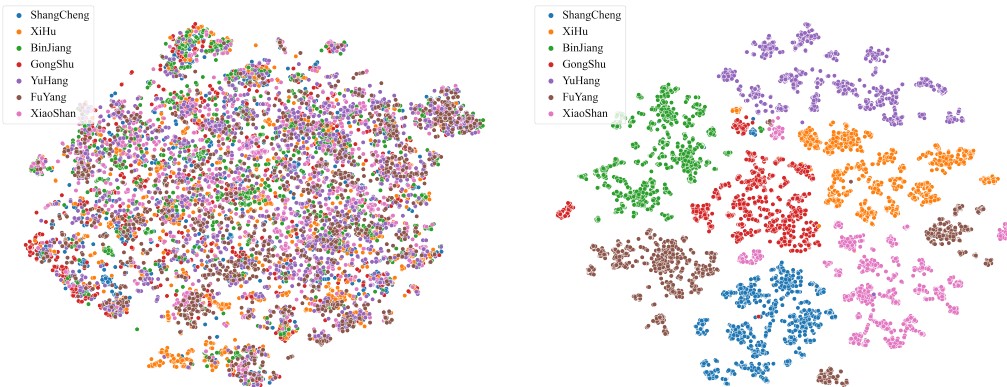

(a) **The t-SNE visualization of embeddings produced by BERT.**   (b) **The t-SNE visualization of embeddings produced by G2PTL.**

Figure 4: The 2D t-SNE projection of 7 districts in Hangzhou City.

Table 5: The sample aggregation evaluation results of the t-SNE visualization results.

| PTM | Silhouette Score ↑ | CH Index ↑ |
|---|---|---|
| BERT | -0.072 | 77.17 |
| MacBERT | -0.042 | 29.33 |
| ERNIE 3.0 | -0.064 | 67.53 |
| GeoBERT | 0.009 | 2227.06 |
| ERNIE-GeoL | -0.054 | 802.24 |
| MGeo | 0.173 | 3772.61 |
| G2PTL | **0.264** | **4,854.81** |
| -w/o GL | -0.006 | 890.07 |
| -w/o GT | -0.049 | 652.76 |
| -w/o HTC | -0.025 | 431.85 |

## B.2 GEOGRAPHICAL KNOWLEDGE OF ADMINISTRATIVE REGIONS

We randomly replace single or multiple geographic entities in address with "[mask]" tokens and then predict them by using PTM. The prediction results can only be selected from geographic entities of the same administrative level, which belong to the upper-level geographic region. We curate two test cases to check the prediction results of PTMs, as shown in Figure 6 and Figure 7. We show the prediction probabilities of the PTM for different geographic entities, and the labels are highlighted in the figure. For Case 1, the prediction results of BERT and G2PTL are correct, but the prediction

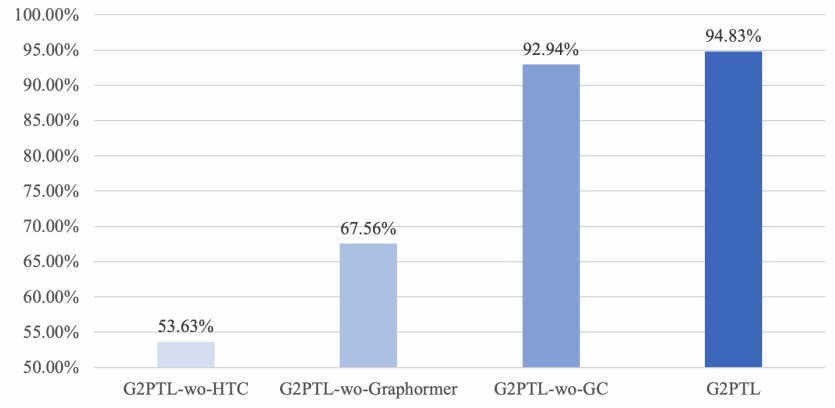

Figure 5: The prediction accuracy of G2PTL for geographic entity.

results of BERT are not as discriminative as the results of G2PTL. For Case 2, the prediction result of BERT is wrong, and the prediction probability of G2PTL for the label is still far ahead. This shows that G2PTL has learned the administrative region information of geographic entities and the relationship between different administrative regions. We randomly masked the geographic entities in 10,000 addresses, and then use them to calculate the prediction accuracy of G2PTL. The experimental results are shown in Figure 5. The prediction accuracy of G2PTL for geographic entities reaches 94.83%. The HTC task brings a 41.2% improvement in absolute accuracy, which shows that the acquisition of administrative knowledge in G2PTL mainly relies on the HTC task.

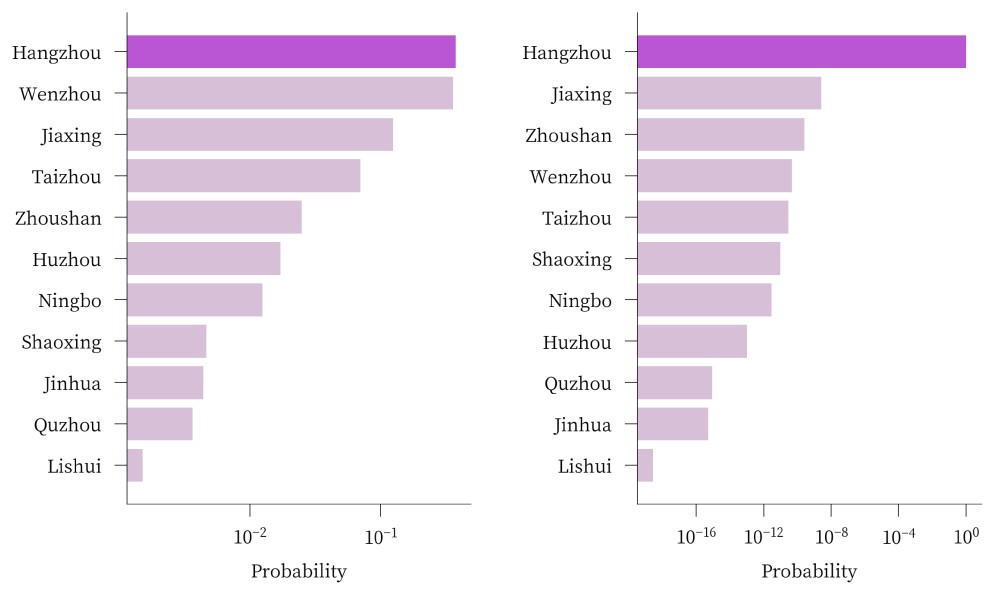

(a) **The prediction results of BERT.**          (b) **The prediction results of G2PTL.**

Figure 6: Case 1. The prediction results of "XiXi Center, No.588, West Wenyi Road, Jiangcun Subdistrict, Xihu District, [MASK], Zhejiang Province".

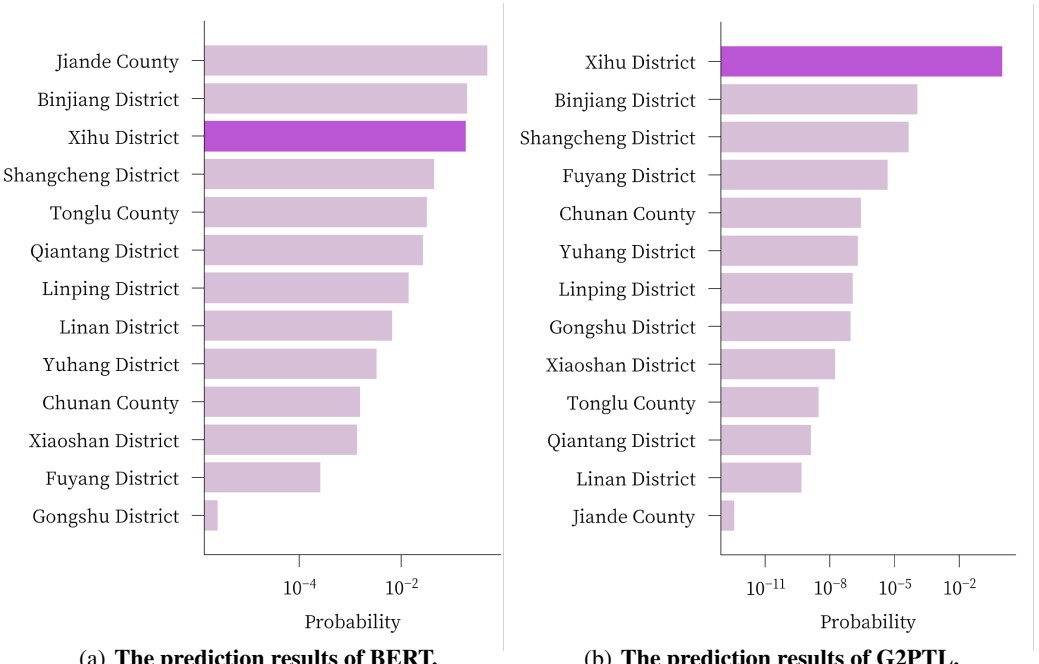

(a) **The prediction results of BERT.**          (b) **The prediction results of G2PTL.**

Figure 7: Case 2. The prediction results of "Hupao Park, Hupao Road, Xihu Subdistrict, [MASK], Hangzhou, Zhejiang Province".

