# OpenReview forum: "G2PTL: A Pre-trained Model for Delivery Address and its Applications in Logistics System"
_ICLR.cc/2024/Conference — Submitted to ICLR 2024_

### Official Review · Reviewer_q3Pr · 2023-10-30

**Soundness:** 3 good
**Presentation:** 3 good
**Contribution:** 3 good
**Rating:** 8
**Confidence:** 5

**Summary:**

In this paper, the authors proposed to build a pre-trained language model for delivery addresses using both text information and geography-graph information. This pre-trained language model, named G2PTL, can improve the performance of downstream tasks such as address entity extraction, address normalization, as well as geolocation coding and pick-up estimation time of arrival. The contributions for this paper are as follows:
1. It pre-trained a large language model for addresses using delivery information. There were similar models in previous literature but this proposed model is optimized for tasks related to logistics and delivery.
2. In pre-training phase, the authors proposed a novel method to use both graphical information as well as text information. In specific, the Graphormer is used to encode both the routing and pairwise distance between addresses, while the Transformer is used to encode the semantic information in address text as well as node degree and position information. Then a new transformer is used to merge two sides of information together.
The authors have shown the performance improvement using this pre-trained language models and they also showed the importance of pre-training tasks such as graph learning and geocoding to the performance improvement of the pre-trained models.

**Strengths:**

The strength of this paper is as below:
1. The proposed pre-trained LM for delivery address is a domain specific language models. Unlike the generic LM, this model focuses on optimizing the performance of logistic and delivery related tasks. For that reason, it is not satisfactory of the semantic text information in the address but also requiring additional knowledge on the relationship between addresses in the geographical map. This information including the neighborhood information from both distance perspective and the routing perspective. Thus, it is natural to consider both of them during the pre-training.
2. The proposed pre-training model for delivery address used multi-modal (text + graph) information in the pre-training phase. Allowing the learned embedding to capture both semantic similarity and the geographical similarity between addresses. This makes a lot of senses since in practice, many similar worded addresses are actually very far away from distance point of view thus is not optimized to be included in the delivery route.
3. The design of heterogenous graph using delivery route as well as AOI Co-locate information is very interesting. It makes sense considering that the delivery system and courier would optimize their route to prioritize addresses in a closer neighborhood and to set the delivery priority accordingly. Building a graph using such information would naturally include geographical neighborhood information which is better than just geo-encoding since a closer direct distance of two addresses may take a long time to travel due to geographical barriers.
4. This paper has well demonstrated the strength of specialized LM in domain specific tasks. In many special domains, pre-training LM is more useful than fine-tuning existing ones given enough resources. The experiments are well written and the result is significant.

**Weaknesses:**

The weakness is listed as below:
1. It is unclear given the graphical information encoded during the pre-training phase, if the proposed model G2PTL can be used to perform graph-related tasks such as link prediction and clustering. The proposed downstream tasks focus on text related tasks only. It is likely due to non-symmetric roles that the graph and the text plays for the design of the model. We would like to see the performance of the graph-related downstream tasks such as link-prediction, node classification based on fine-tuning of this model.
2. This paper used less than 100K samples in the pre-training. It is unclear how the performance of the model scale with the sample size. Since both graph and text are used in the training, it is interesting to know if the model can still train with just text information
3. The inference for the model may need to have completed graph. This is a limitation for its widely use since most of applications do not have a complete graph at hands for inference.

**Questions:**

1. At inference time, do I need to have graph as input besides the text input for this model? From the model design, it seems that the graphomer is used in the model, thus I would expect edge and node input for the model? If the task is just address normalization without graph input, can we still use this model?
2. Do we need to have a correct geographical map before using the proposed model in real life? What if the map is incorrect? How robustness of this model when the graphical information in inference time is noisy?
3. How large the graph do we need to prepare before using this model for address normalization? Do we need to have a global map or local map? If only local maps are used, is it possible that this method cannot learn the address similarity beyond a local region?

---

> ### Author Response · Authors · 2023-11-13
> **Author Response to Reviewer q3Pr**
>
> Dear reviewer, thank you for your valuable suggestions on our work. Below are our detailed responses to your questions:
>
> **Questions 1: At inference time, do I need to have graph as input besides the text input for this model? From the model design, it seems that the graphomer is used in the model, thus I would expect edge and node input for the model? If the task is just address normalization without graph input, can we still use this model?**
>
> **Response：**
>
> During the model inference phase, addresses need to be constructed as a graph. For tasks such as address normalization without graph input: Just build a single-node subgraph based on the address (This is a Self-Loop subgraph, edge_type_code = 1000 (Have a self-loop, no delivery route, no AOI co-locate, and no Alias)), and then enter this subgraph into the model.
>
> Because when sampling subgraphs during the training phase, a weighted adjacency matrix is used to describe the relationship between nodes, and the weight value is edge_type_code. In particular, we use self-loop edge to describe the relationship between the node and itself. Its edge_type_code is 1000, which is used to distinguish it from the other 3 types. Therefore, when the downstream task does not have a graph input, you only need to build a single-node graph and add a self-loop edge to this node, which can be used for fine-tuning and inference of a single address. (This part of the code is also in the open source link:https://huggingface.co/Cainiao-AI/G2PTL. All experiments in the paper are tested in this way.)
>
> **Questions 2: Do we need to have a correct geographical map before using the proposed model in real life? What if the map is incorrect? How robustness of this model when the graphical information in inference time is noisy?**
>
> **Response：**
>
> 1. Whether a geographical map is required depends on your application scene.
>
>     ○ If your application scene is a graph-level task (such as graph-based ETA, path prediction, etc.), we recommend that you input correct geographical map information to the model, which will help the model make more accurate judgments. If you do not have a geographical map, you can construct S2 co-locate edges through the S2 method to replace the AOI co-locate edges (this edge can also represent part of the geographical co-location information, but the method of dividing the earth is different), but the lack of other edges will reduce the performance of the model.
>
>     ○ If your application scene is a single-address level task (such as address normalization, geocoding, etc.), at this time, you only need to construct the address as a single-node graph(The method is in the response to question 1).
> 2. When creating large-scale heterogeneous graphs, noise is already present in it ：
>
>     ○ AOI co-locate：Limited by the performance of the geocoding service, the data source of AOI co-location edge may be wrong
>
>     ○ Delivery Route: the courier’s delivery route may be wrong (positioning drift, etc.)
>
>     ○ Alias: As shown in Fig1, each node will have multiple address descriptions, and the user will bring in input noise when writing the address.
>
> We believe that these noises will slowly disappear as the amount of training data increases, because these noises are unbiased. At the same time, these noises will enhance the adaptability and robustness of the model. This is also reflected in our actual industrial deployment applications. We have deployed G2PTL into our online logistics system, which processes tens of millions of addresses every day (with the development of logistics business, there are many a new address that the model has never seen before). After the deployment of G2PTL, the accuracy of the address entity recognition online service has been improved by about 9% (84%->93%, the previous online model was BERT), which also reflects the good anti-noise performance of the model.
>
> **Questions 3: How large the graph do we need to prepare before using this model for address normalization? Do we need to have a global map or local map? If only local maps are used, is it possible that this method cannot learn the address similarity beyond a local region?**
>
> **Response：**
>
> If you want to use this model for address normalization, you do not need to prepare a map, but you need to use the addresses to build a single-node graph as input (The method is in the response to question 1).
>
> In the inference phase, the model only needs a local map; in the training phase, a global map is needed.
>
> If you use a local map, our method can only learn information such as the spatial relationship of addresses inside the local map, but cannot learn the spatial relationship between addresses outside the local map. For addresses outside the local map, the model can only calculate the similarity of text-level addresses because the model has not learned the spatial information of the corresponding address.

---

### Official Review · Reviewer_VFda · 2023-10-31

**Soundness:** 2 fair
**Presentation:** 2 fair
**Contribution:** 2 fair
**Rating:** 5
**Confidence:** 4

**Summary:**

This paper builds a pre-trained graph model G2PTL for the logistics domain and applies it to downstream tasks. The authors first process the delivery data and use it to construct a large-scale heterogeneous graph. To pre-train G2PTL, they propose three pre-training tasks: whole word mask, geocoding, and hierarchical text classification. Finally, the authors validate the effectiveness of G2PTL on four different types of downstream tasks. The main contribution of this paper is to propose a paradigm for constructing pre-trained models for the logistics domain.

**Strengths:**

1. The proposed graph construction method is novel, providing a good reference for pre-training methods in the logistics domain.

2. The paper is clearly written and easy to read and understand.

3. The ablation experiments and performance on downstream tasks demonstrate the effectiveness of the proposed G2PTL.

**Weaknesses:**

1. The proposed method is a combination of existing technologies, such as whole word mask, geocoding, and Graphormer.

2. Missing discussions on necessary details, such as inference efficiency, data distribution of pre-training tasks, convergence analysis of pre-training, parameter selection, and optimizing strategies.

3. Missing statistical significance tests.

**Questions:**

1. Given that a fixed heterogeneous graph has already been predefined, how can the proposed method be scaled to new addresses?

2. How do you balance the significance of various tasks within the loss function? Based on the findings from the ablation experiments, it appears that HTC plays a more crucial role in pre-training.

3. What is the rationale behind choosing to sample from the entire graph as opposed to creating subgraph-level graph models? The former option demands significantly greater computational and storage resources.

---

> ### Author Response · Authors · 2023-11-14
> **Author Response to Reviewer VFda**
>
> Dear reviewer, thank you for your valuable suggestions on our work. Below are our detailed responses to your questions:
> **Questions 1: Given that a fixed heterogeneous graph has already been predefined, how can the proposed method be scaled to new addresses?**
>
> **Response：**
>
> 1. If you are only using the model for inference on new addresses, you only need to build a single-node subgraph based on the new address(This is a Self-Loop subgraph, edge_type_code = 1000 (Have a self-loop, no delivery route, no AOI co-locate, and no Alias)), and then enter this subgraph into the model. Because when sampling subgraphs during the training phase, a weighted adjacency matrix is used to describe the relationship between nodes, and the weight value is edge_type_code. In particular, we use self-loop edge to describe the relationship between the node and itself. Its edge_type_code is 1000, which is used to distinguish it from the other 3 types. Therefore, when a downstream task takes a new address as input, you only need to build a single-node graph and add a self-loop edge to this node, which can be used for fine-tuning and inference of a single address. (This part of the code is also in the open source link:https://huggingface.co/Cainiao-AI/G2PTL. All experiments in the paper are tested in this way.)
>
> 2. If you want to add new addresses to the model for fine-tuning. There are 2 ways to extend the model to new addresses:
>
>     ○ If there are a large number of addresses, a new heterogeneous graph can be constructed and trained according to the pre-training method we proposed. (If you cannot obtain edges with domain knowledge such as AOI co-locate and Delivery Route, you can use edges similar to S2 co-locate (refer to ERNIE-GeoL，KDD‘22))
>
>     ○ If there are only a small number of addresses (<=10k), we recommend fine-tuning the model using single-node subgraphs constructed from new addresses.
>
> **Questions 2: How do you balance the significance of various tasks within the loss function? Based on the findings from the ablation experiments, it appears that HTC plays a more crucial role in pre-training?**
>
> **Response：**
>
> We have observed that the convergence speeds of different losses are different during the training process. Among them, HTC-Loss converges the fastest because this task is simpler than MLM and Geocoding tasks (HTC only needs to focus on administrative hierarchy relationships among address entities. It only needs to learn all geographical administrative relationships, and the solution space is smaller than the other two tasks). We did not do any special work to adjust the balance between the three loss functions. We just maintained the loss weight setting of 1:1:1. Under this setting, we trained for about 1 week.
>
> Due to training time and cost reasons, we terminated the training. At this time, HTC task has converged, MLM task is close to convergence, and the GC task has not fully converged. Our training loss log is also in the open source link:https://huggingface.co/Cainiao-AI/G2PTL.
>
> We believe that different tasks focus on different knowledge.
>
> In the experiment in Table 1, both the AEP and AET tasks are for a single address, and they should pay more attention to the the relationship of sub-words in the address. Therefore, from the experimental results, the importance of the HTC task will be higher, which is consistent with the goal of HTC task design.
>
> For tasks related to geographical knowledge (absolute position positioning of addresses and relative positions between addresses), the importance of GC will be higher (the ablation experiments of the GC task in Table 1 and the ablation experiments of table 5 in the Appendix both prove this).
>
> The MLM task helps the model handle more input noise (incomplete, partially duplicated, misspelled, etc.) and ensures the model's ability to understand address semantics. The cases in Figures 6 and 7 also prove the effectiveness of the MLM task.

---

> ### Author Response · Authors · 2023-11-14
> **Author Response to Reviewer VFda**
>
> **Questions 3: What is the rationale behind choosing to sample from the entire graph as opposed to creating subgraph-level graph models? The former option demands significantly greater computational and storage resources.**
>
> **Response：**
>
> We choose to sample from the entire graph for the following reasons:
> 1. Our goal is to train a PTM that contains rich delivery knowledge and geographical knowledge and can be applied to geographically related downstream tasks. Therefore, we believe that the training samples of the model should contain a large amount of delivery knowledge and geospatial knowledge. Subgraph-level models may not capture the global structure and relationships present in the entire graph. They may miss out on long-range dependencies and global patterns that can be crucial for certain tasks. Using a subgraph-level graph models will result in the loss of information between nodes, which is detrimental to training a PTM.
> 2. In our entire graph, the distribution of edge types is nonuniform. Take AOI co-locate as an example: This edge type will only appear between different nodes within the same AOI. It can only describe the co-locate relationship of nodes within the same AOI, and is a strong local pattern. It cannot describe the relationship between node pairs between two AOIs (even if they are adjacent in real geographical space, such as Node 1 and Node 2 in Fig. 1). If we only use AOI co-locate edges to build a sub-graph model, it may only model the local geographical knowledge in our system and miss the global geographical information. Delivery Route edges can supplement it with the geographical information of Cross-AOI, making the edge distribution more consistent with the real world(a closer direct distance of two addresses may take a long time to travel due to geographical barriers.). Details of our heterogeneous graph and the distribution of edges in the heterogeneous graph can be found in the Appendix of the paper.

---

> > ### Comment · Reviewer_VFda · 2023-11-22
> > **Response read**
> >
> > I have read the response to my questions and deciced to keep my rating.

---

### Official Review · Reviewer_PNNu · 2023-11-06

**Soundness:** 3 good
**Presentation:** 3 good
**Contribution:** 3 good
**Rating:** 6
**Confidence:** 4

**Summary:**

This work introduces a novel pre-trained model designed specifically for delivery addresses in logistics tasks, named G2PTL.
Central to G2PTL's innovation is its unique architecture that leverages graph-based representations of address data.
This architecture supports the model's capability to efficiently learn geographic knowledge and delivery details through
three distinct pre-training tasks: Masked Language Modeling (MLM), Geocoding, and hierarchical text classification.
A distinguishing feature of G2PTL is its adeptness in modeling graph information inherent in the logistics domain.

**Strengths:**

* G2PTL's architecture adeptly captures diverse and complex real-world delivery information in the form of heterogeneous graph.
* The strategy of employing subgraphs sourced from a larger heterogeneous graph for training is innovative.
* G2PTL demonstrates strong performance across a suite of logistics-specific tasks, such as Geocoding, ETA for Pick-up, address entity prediction, and address entity tokenization.
* The work is complemented by a thorough analytical review.

**Weaknesses:**

* The representation in Figure 1 lacks clarity. The relationship between the left and right sections of the figure is puzzling.
For instance, the right side depicts an edge between node 1 and node 2 labeled "001", suggesting "no delivery route, no AOI co-location, and has Alias." However, the left side appears to contradict this, showing a delivery route between node 1 and node 2.

**Questions:**

Please respond to the weaknesses I listed.

---

> ### Author Response · Authors · 2023-11-13
> **Author Response to Reviewer PNNu**
>
> Dear reviewer, thank you for your valuable suggestions on our work. Below are our detailed responses to your questions:
> **Questions 1: The representation in Figure 1 lacks clarity. The relationship between the left and right sections of the figure is puzzling. For instance, the right side depicts an edge between node 1 and node 2 labeled "001", suggesting "no delivery route, no AOI co-location, and has Alias." However, the left side appears to contradict this, showing a delivery route between node 1 and node 2.**
>
> **Response：**
>
> Sorry, we didn't describe the edge encoding clearly enough. In the binary encoding result of the edge, position 0 is the last bit in the result(rightmost) and position 2 is the first bit(leftmost). Therefore, the edge labeled "001" between node 1 and node 2 in  Figure 1 indicates " no AOI co-locate, no alias, only delivery route".
>
> | code position | position 2 | position 1 | position 0 |
> | :--: | :--: | :--: | :--: |
> | edge type description | alias | AOI co-locate | delivery route |
> | edge type code | 0 (None) | 0 (None) | 1 (Have) |

---

> ### Comment · Reviewer_PNNu · 2023-11-23
> **keep my rating.**
>
> I have read the authors' responses and decided to keep my rating.

---

### Public Comment · ~Dongjiang_Cao1 · 2023-11-27
**How to test with only a single address**

During training, the input consists of information in the form of a graph structure. However, during testing, is it possible to input only a single address without constructing a graph? How can this be implemented?

---

### Meta-Review · Area_Chair_Bm1w · 2023-12-07

**Metareview:**

This paper presents a domain-specific pre-trained model called G2PTL for delivery address in Logistics field. It utilizes real-world logistics delivery data to construct a large-scale heterogeneous graph of delivery addresses, and then pre-trains the model with subgraphs sampled from the heterogeneous graph. G2PTL can improve the performance of delivery-related downstream tasks.

The proposed G2PTL model is reasonable, and it outperforms baseline models on delivery-related downstream tasks. The paper is easy to follow. The dataset and the graph construction method are interesting, but the innovation of the model architecture and pre-training tasks is limited. The proposed model is only trained and used in a very narrow domain - delivery address, and it is unclear about the model's generalization ability to other domains. Reviewers also suggest to test on additional graph-related downstream tasks such as link-prediction and node classification.

This paper is a borderline case and I am leaning towards weakly rejecting it.

**Justification For Why Not Higher Score:**

see the meta-review.

**Justification For Why Not Lower Score:**

N/A

---

### Decision · Program_Chairs · 2024-01-16

Reject